# Preparation and Characterization of the Flame Retardant Decorated Plywood Based on the Intumescent Flame Retardant Adhesive

**DOI:** 10.3390/ma13030676

**Published:** 2020-02-03

**Authors:** Muting Wu, Wei Song, Yuzhang Wu, Wei Qu

**Affiliations:** Research Institute of Wood Industry, Chinese Academy of Forestry, Beijing 100091, China; wmt1996@caf.ac.cn (M.W.); sw@caf.ac.cn (W.S.); wyz@caf.ac.cn (Y.W.)

**Keywords:** flame-retardant plywood, decorated plywood, intumescent flame retardant, cone calorimeter, single burning item

## Abstract

A novel type of flame-retardant decorated plywood (FDP) was designed and prepared based on one kind of intumescent flame-retardant adhesive. The flame-retardant adhesive was composed of the phosphorous-nitrogen flame retardant and melamine urea formaldehyde (MUF) resin. An adhesive was placed between the plywood substrate and the decorative veneer. The shear strength of the FDP satisfied the Class II (GB/T 9846) when the ratio of flame-retardant and MUF was less than 0.65. The thermal stability of the flame-retardant adhesive was measured by thermogravimetric analysis (TGA). The intumescent behaviors of adhesives were systematically investigated. The morphological and chemical structures of the intumescent char of the flame-retardant adhesive were characterized by the scanning electron microscopy (SEM) and Fourier transform infrared spectra (FTIR), respectively. The fire performance of FDP was assessed by the cone calorimeter and the single burning item test. The FDP exhibited the most effective barrier when the optimized ratio of ammonium polyphosphate (APP) and pentaerythritol (PER) in the adhesive is 3. The flame-retardant class of FDP could be up to class B1(B) according to GB/T 8624.

## 1. Introduction

Plywood is one of the main materials used for construction and for furniture. However, the flammability of plywood has limited its application in public places or high-rise buildings, especially after the implementation of Chinese National Standard GB 20286-2006 [1]. The methods used to obtain the flame retardant plywood have been studied extensively. Three approaches are commonly employed to provide wood-based products with improved fire retardant performances: flame-retardant coatings [2,3]; veneer or board chemical impregnation [4]; and physical mixing of fire retardant into the adhesives [5,6]. Although the fire retardant coating is able to reduce the flammability of wood products, the aging and cracking of the coating are still a challenging problem [7]. An aqueous solution of fire retardant is almost necessary for the impregnation of veneer or plywood, which often leads to the leaching of fire retardant [8]. It was found that the mechanical strength values of panels produced by using the veneers treated with fire retardant chemicals were lower than those of control panels [9]. The use of fire retardant glue is a convenient way to manufacture the flame retardant wood-based panels, but it should be discussed whether the flame retardant is added into every glue layer [10]. It is nearly useless that there is a flame retardant in the middle layer of plywood at the first stage of fire. The flame retardant is dispersed in the whole panel, which leads to a flame retardant efficiency decrease and a cost increase. A novel flame retardant method for wood-based panels is required to avoid the above shortcomings of the traditional methods.

Decorated panels are mainly made from the common panels and the covering materials that can be paints, veneer, laminating plastics or impregnated papers. Many laminated wood panels are covered with the resin impregnated papers [11]. The decorated panels are the main materials of custom-made furniture. Statistical results show that more than 90% of medium density fiberboard (MDF) and particleboard (PB), and 60% of plywood are overlaid with the impregnated paper or the decorative veneer in China [12]. If the substrates of the decorated panels were flame retardant, then the flame retardant and decorative functions are combined [13]. The application of decorated panels will expand into the engineering field, for example, the wall cover materials of the light wood constructions. The furniture made from the flame retardant decorated panels can keep combustible goods away from fire, and can establish fire isolation zones [14].

Intumescent flame-retardant (IFR) has gained much attention due to its lower cost and better performance than other retardants [15]. In the event of a fire, the IFR expands on contact with heat to provide a thermally insulating char that delays the diffusion of heat to substrates. The flame-retardant performance of the IFR is influenced largely by the matrix resins [16]. The polyacrylate, epoxy and melamine-formaldehyde resins (MF) have been widely investigated in the commercial intumescent coatings [17,18]. MF resin works as both the matrix and the blowing agent. Compared with MF, melamine urea formaldehyde resin (MUF) has a lower cost and longer store stability, which is important for the wood industry [19]. Hu and co-workers [20] firstly mixed MUF with IFR and then coated it on the surface of pre-expanded polystyrene particles to prepare flame retardant expandable PS foams. The mixture of APP and MUF was coated on the surface of medium density fiberboard by Wu and co-workers [21]. However, a relationship has not been established between the quality of the char and the fire performance of the mixture of MUF and IFR.

In this study, the flame retardant decorated plywood (FDP) was prepared based on the intumescent flame retardant adhesive. The flame retardant layer was established between the decorative layer and plywood. The relationship between the structure of char and the flame retardant performance was studied systematically. The thermal stability of the flame retardant resin was investigated by using thermogravimetric analysis (TGA). The intumescent behaviors of adhesives were investigated by using a scanning electron microscope (SEM) and Fourier transform infrared spectroscopy (FTIR). The flame-retardant properties of FDP were measured by using a cone calorimeter and the single burning item (SBI) test.

## 2. Materials and Methods

### 2.1. Materials

Ammonium polyphosphate (APP, n > 1000, Mn > 208,000) was purchased from Sichuan Changfeng Chemical Co., Ltd. (Deyang, China). Pentaerythritol (PER), melamine, urea and formaldehyde solution (37–40%) were purchased from Beijing Modern Eastern Fine Chemical Co., Ltd. (Beijing, China). Sodium hydroxide (NaOH) was acquired from Sinopharm Chemical Reagent Co., Ltd. (Shanghai, China). Decorative veneers and plywood were supplied by Shijiazhuang Huajie Wood Co., Ltd. (Shijiazhuang, China). All materials were used as received.

### 2.2. Sample Preparation

MUF resin was prepared with formaldehyde, melamine and urea [14]. First, 120 g of formaldehyde solution (37%) and 40 g of urea were mixed in a reactor equipped with thermometer and mechanical stirrer, and the pH was adjusted to 8.0–9.0 with 1 M NaOH. The mixture was heated to 90 °C for 35 min. Then the pH was adjusted to 5.0–5.5 with NH4Cl, followed by the addition of 15 g of urea and 10 g of melamine for 40 min with the pH = 8.5. MUF resin was successfully synthesized and cooled down. The molar ratio of formaldehyde to urea and melamine, F/(M+U), was 1.3. The flame retardant adhesive was prepared by mixing the APP, PER and MUF, which were then fully stirred by the rotary agitator. The formulations of the intumescent flame retardant adhesive are listed in Table 1.

The structure of the FRD-plywood is shown in Figure 1. The flame retardant layer was established between the decorative layer and the substrate. The flame retardant layer was composed of APP, PER and MUF. The decorated plywood was prepared through two steps. Firstly, the flame retardant adhesive resin was coated on the surface of poplar plywood (300 mm × 300 mm × 9 mm). Secondly, a piece of decorative veneer (310 mm × 310 mm × 0.5 mm) was hot-pressed on the top of the coated plywood with 8 MPa and at 110 °C, and was maintained for 5 min. The specimen for CONE was cut into 100 mm × 100 mm pieces. FRD-plywood (a big scale sample for SBI) was prepared using A3P1 adhesive and plywood (1220 mm × 2440 mm × 18 mm).

The flame retardant steel boards (for back temperature tests) were prepared by coating the resin on the steel boards (100 mm × 100 mm × 2 mm). The char samples of the flame retardant adhesive were prepared in muffle furnace for 30 s at 800 °C.

### 2.3. Measurements and Characterization

The wet shear strength of plywood samples was measured in accordance with the Chinese National Standard (GB/T17657-1999). Twelve plywood specimens (2.5 cm × 10 cm) cut from two plywood panels were submerged in water at 63 ± 2 °C for 3 h, and then dried at room temperature for 10 min before tension testing. The bonding strength was calculated using following equation:Bonding strength (MPa) = tension force (N)/gluing area (m^2^)

Thermogravimetric analysis (TG) was performed by using a SDT Q600 thermal analyzer (TA Instruments, New Castle, DE, USA). The sample with a 3 mg weight was placed in an alumina crucible and at a heating rate of 10 °C /min under an N2 atmosphere from 30 to 800 °C.

Char volume was measured by paraffin embedding to reinforce the char structure and insulate the water. The char morphology was monitored by using a digital camera (Sony, Tokyo, Japan).

The micromorphology images of char were analyzed by using scanning electron microscopy (SEM, S4800, Hitachi, Tokyo, Japan) at high vacuum conditions with a voltage of 10 kV. Prior to observation, all samples were gold-sputtered to improve their conductivity.

Fourier transform infrared spectra (FTIR) of resin and its char were recorded by using Nicolet Is50 (Thermofisher, Waltham, MA, USA) from 400 to 4000 cm^−1^. The differentiation rate was 4 cm^−1^ and the number of scans was 8.

Laser Raman spectroscopy (LRS) was carried out on a micro Raman imaging spectrometer (DXRxi, Thermofisher, Waltham, MA, USA) with a 532 nm helium-neon laser line. The laser beam focused on the surface to scan in the 50-3400 cm^−1^ region.

Cone Calorimeter (FTT UK, Derby, UK) was used to evaluate the flammability of plywood based on ISO 5660-1 [22] under an external heat flux of 50 kW/m^2^. Three replicates were tested were calculated for each plywood type. The Back temperature was measured by using thermocouples (Testo 176 T4, Testo, Lenzkirch, Germany) laid on the bottom of steel board test under CONE from 0 to 700 s under the external heat flux of 50 kW/m^2^.

Single burning item (SBI, Motis, Kunshan, China) was used to evaluate the combustion results of plywood according to the China National Standard (GB/T 20284 and GB/T 8626) from 0 s to 1300 s. The same method was used to classify most building products in Europe (EN 13823). The specimen was mounted on a trolley that is positioned in a frame beneath an exhaust system. The reaction of the specimen to the burner was monitored instrumentally and visually. Heat and smoke release rates were measured instrumentally and physical characteristics were assessed through observation. According to GB/T 8624, the building products were assessed into classes A1, A2, B1, B2, and B3, which are different from the classes A-F in Europe. For example, the classes B and C in the European criteria belong to class B1 in GB/T 8624, and were written as B1(B) and B1(C), respectively. The test results were required for a classification A2 (combined with the test results from GB/T 5464 and GB/T 14402), B1and B2 (combined with test results from GB/T 8626). These were given in Table 2. FIGRA is an indication of fire growth rate based on heat release. LFS indicates whether the spreading of a lateral flame to the end of the long wing was observed or not.

## 3. Results and Discussion

### 3.1. Shear Strength Test

Shear strength is a key parameter for adhesives. When flours are added as fillers for an adhesive, the rheological behavior and the curing properties of the adhesive are changed [23]. Previous studies have shown that the flame retardant containing phosphorus often leads to a bonding strength reduction [24]. In this work, wet shear strength of the plywood was evaluated using various ratios of MUF and IFR. As shown in Table 3, the wet shear strength decreased with the increased addition of IFR (APP/PER = 3, m/m). The shear strength of plywood is more than 0.97 MPa when the ratio of IFR to MUF is less than 0.60, which satisfies the requirement of the China National Standard about plywood (GB/T 9846, 0.7 MPa). The discussion of the flame-retardant properties of FDP should be based on having enough shear strength under the condition of the ratios of MUF and IFR being more than 100:60.

### 3.2. Thermal Degradation Behavior of the Adhesive

The TGA curves of the adhesives in inert atmosphere are shown in Figure 2. The related data are listed in Table 4. The curves can be divided into two groups. The curves of A1P2, A1P3 and MPER are below that of A1P1, A2P1, A3P1 and MAPP, which is attributed to the amount of PER in the adhesive. With the higher addition of PER in the adhesives, more weight loss is obtained at the same temperature. The temperature associated with a weight loss of 5% is indicated as T-5%. MPER has the lowest T-5%, and A1P1 is the watershed of T-5%. The T-5% of A3P1, A2P1, and A1P1 are circa 183 ^o^C, while the T-5% of A1P2 and A1P3 are circa 170 °C. Compared with the A3P1, the T-5% of MAPP is decreased without PER. The residue at 800 °C of samples has a similar trend, indicating that APP and PER cooperate to prevent further degradation of the adhesive, and that the ratio of APP and PER is a key parameter for the thermal stability of adhesives [16].

### 3.3. Intumescent Behaviors of Adhesives

The intumescent behaviors of adhesives are closely related to their flame-retardant performance. The expansion volume or height of intumescent char is one key parameter of the IFR coating. The maximum expansion volumes are measured as an index to evaluate the IFR systems [25,26]. The expansion volume of adhesives is measured by the drainage method with paraffin embedding. The char of adhesives is prepared at 800 °C for 30 s in the muffle furnace with the coating weight of 333 g/m^2^. The images of the intumescent char are shown in Figure 3 and the data of expansion volume (ex-V) is listed in Table 5. The ex-V of both MAAP and MPER is not high, which indicates that it is necessary to mix APP and PER for intumescent flame retardant systems. The ex-V of the mixtures of APP and PER is increased as the proportion of APP increases. A3P1 has the maximum expansion volume, 133 mL, which is more than two times that of A1P3. The intumescent process is triggered by APP releasing acid in the event of a fire. The acid reacts with PER, resulting in its dehydration and forming a carbonaceous char. The blowing agent, melamine in MUF, decomposes and releases gases in the molten mass, resulting in foaming which then solidifies, making a heat transfer barrier [27]. When the ratio of APP and PER is 3, the viscosity of molten mass is the most suitable for expansion.

The SEM test gives some information on the changes in the morphological structure of intumescent char. The thermal conductivity of intumescent char depends on the cell structure of char [28]. The SEM micrographs of char of adhesives are presented in Figure 4. The char of MAPP can form a honey bomb structure, which can effectively insulate the heat transfer but still with cracks [29]. MUF also worked as the carbon source without PER. The ex-V of MAPP and the cracks of char indicate that the viscosity of the ester formed by APP and MUF is not high enough for expansion. The heat can be transferred by the crack between the pores. Although there were some open pores in the char of A3P1, the char is continuous without any fractures. Some micron bubbles are on the surface of the char, which can improve its heat insulation [30]. The images of char of A1P1 and A1P3 show that the char was expanded completely and the thickness of cell wall was very low. But the cell was broken and connected, which was not good for heat insulation. The obvious bubble aggregation can be ascribed to the PER’s lack of dehydration for APP.

### 3.4. FTIR of Adhesives

Figure 5 shows FTIR spectra of the adhesive, its components and its char. The absorption peaks of MUF at 1336 cm^−1^ and 1554 cm^−1^ were attributed to the ring vibration of melamine in the MUF resin, C–N and C=N stretching vibration [31]. The peak of MUF at 1658cm^−1^ was attributed to C=O stretching vibration [32]. The absorption peaks of APP were at 800 cm^−1^, 874 cm^−1^, 1011cm^−1^, 1253 cm^−1^ and 1445 cm^−1^, which were attributed to PO2, PO3 and P–O bending vibration, P=O, P–O symmetric stretching vibration, and N–H bending vibration, respectively [33]. The peak of PER at 2950 cm^−1^ was the typical absorption of a (C–H) bond. Comparisons of the FTIR spectra of MUF, APP, PER and A3P1 showed that no new absorption peaks were present in the FTIR spectra of A3P1 and the characteristic peaks of MUF, APP and PER also existed in the curve of A3P1. After char formation, the new peaks of the char of A3P1 (A3P1c) at 975 cm^−1^, 1080 cm^−1,^ and 1660 cm^−1^ were due to the formation of P–N, P–O–C and C=O [34]. The disappearance of the peaks at 874 cm^−1^, 1250 cm^−1^, 1554 cm^−1^ and 2950 cm^−1^ indicated the degradation of APP, PER and MUF. The chemical structure of the char with different ratios of APP and PER was compared in Figure 5b. The appearance of the peak of A1P1c and A1P3c at 1550 cm^−1^ indicated that the melamine ring had not decomposed completely. The typical absorption of APP still appeared in the curves of MAPPc, which verified that not enough charring occurred without PER.

### 3.5. Cone Calorimeter Measurements

The fire performance of the decorated plywood (DP) and FDP was measured by using a cone calorimeter. The heat release rate (HRR) and total heat release (THR) are the most important parameters for evaluating the reaction-to-fire performance of materials [35]. The HRR and THR curves of one representative sample for each plywood type are shown in Figure 6. For each plywood type, the ignition time (IT), peak heat release rate (PHRR), time to PHRR (TTP), average heat release rate (AHRR) and THR in 180 s and 300 s are summarized in Table 6. Figure 6a shows the HRR curves of DP and part of FDP. DP displayed three distinct PHRRs. The first one is associated with the decorative veneer. The second and third ones are ascribed to the substrate of plywood. After the third PHRR, the HRR of DP decreased monotonically until flame-out occurred. The first and the second PHRRs of DP could be distinguished because of the MUF layer between the decorative veneer and the plywood. As for FDP, the glue layer was taken place by the flame-retardant adhesive. The intumescent char was formed after the delamination of the decorative veneer. The HRR of FDP almost decreased to zero after the first PHRR with the protection of intumescent char. The appearance of the second PHRR of FDP implies the failure of intumescent char. As can be seen in Table 6, the distance between the first PHRR and second one of FDP increased as the proportion of APP increased, apart from in the case of MAPP. A3P1 had the longest time to the second PHRR, indicating the properties of the flame-retardant adhesives are best when the ratio of APP and PER is 3:1. Compared with DP, the AHRR and THR in 300 s of A3P1 decreased by 89.3% and 88.6%, respectively. From 26 s to 15 s, the ignition time decreased with an increased proportion of PER. The reason for this might be that the more flammable volatile gases were produced with the more PER under heat [36].

### 3.6. Single Burning Item Test

The scale of fire tests plays a major role in fire science. The ingle burning item test is one of the most important test methods available and it can be interpreted as a form of large scale testing [37]. Large-scale FDP (2440 mm × 1440 mm × 18 mm) was made for the industrial production lines with the A3P1 adhesive. DP and FDP were tested in a single burning item test and the main results—the fire growth rate and total heat release over 600 s—were used to describe the test results (Figure 7). DP displayed multiple peaks in its curve of HRR. Each of the peaks was associated with the combustion and delamination of a veneer layer in the eleven-ply sample [6]. However, there was no peak in the HRR curve of FDP and the curve was below that of DP before 1050 s of burning. The peak of the HRR curve of FDP in 1050 s indicates the failure of the intumescent char. As shown in Figure 7b and Table 7, the THR600s of DP was 14.9 MJ while the THR600s of FDP was 4.3 MJ, a 71.1% decrease. The total heat release of flame retardant veneer plywood also decreased by 71.1% compared with the result for veneer plywood. The images of the residues of DP and FDP after the SBI test are shown in Figure 8. After 1300 s, the burned area of DP was larger than that of FDP and DP was burned through, which indicates that the pyrolyzing zone of FDP was limited by the flame-retardant adhesive. Based on the images of the cross section of samples, DP had been charred completely, while the substrate of FDP had not started to char. This outcome shows increased fire resistance due to the intumescent char of the flame-retardant adhesive blocking fire penetration in plywood. According to the criteria for compliance in Table 2, DP and FDP are assessed into classes B2(D) and B1(B), respectively. The class B1(B) is the highest level for flame-retardant wood, and satisfies most of the requirements of flame-retardant wood materials.

## 4. Conclusions

In this work, a method to prepare a kind of flame-retardant decorated plywood was established based on the intumescent flame-retardant adhesive. The shear strength of FDP was more than 0.97 MPa when the ratio of IFR was less than 0.60, which satisfies the requirement of the China National Standard around plywood (GB/T 9846, 0.7 MPa). The composition of the flame-retardant adhesive was optimized according to the thermal stability and intumescent behaviors of adhesives. A3P1 had the maximum residue ratio and expansion volume. The morphological structure of the char of A3P1 was a multi-bubble structure, which could improve the heat insulation. The chemical changes of the adhesives in the charring process were characterized by using FTIR. The new bonds of P–N, P–O–C, and C=O were formed after charring, which was related to the ratio of APP and PER. The fire performance of DP and FDP was compared by using a cone calorimeter measurement and a single burning item test. The time to the second peak of FDP was delayed with the proportion of APP increasing apart from MAPP. Compared with DP, the AHRR and THR after 300 s of A3P1 decreased by 89.3% and 88.6%, respectively. The large scale FDP was prepared and measured by using the SBI test. The THR600s of FDP was 4.3 MJ, which was a decrease by 71.1% compared to DP. The flame-retardant class of FDP with A3P1 was up to class B1(B). In conclusion, the novel method used to produce the FDP proved to be feasible. Compared to the traditional methods, the process of using this method is easier and more successfully avoids of the disadvantages of impregnation, physical mixing and flame-retardant coatings.

## Figures and Tables

**Figure 1 materials-13-00676-f001:**
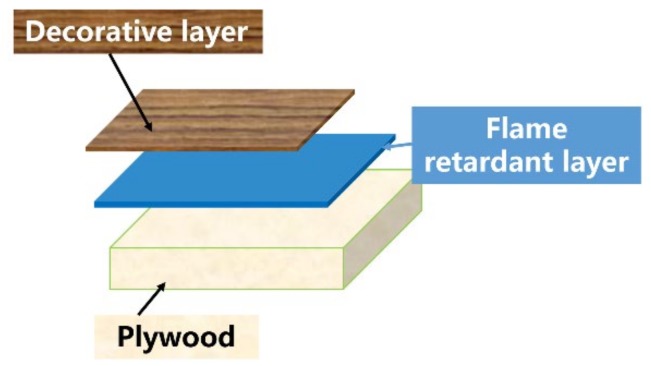
Schematic of the flame retardant veneered plywood. MUF, Melamine-Urea-Formaldehyde; APP, ammonium polyphosphate; PER, pentaerythritol.

**Figure 2 materials-13-00676-f002:**
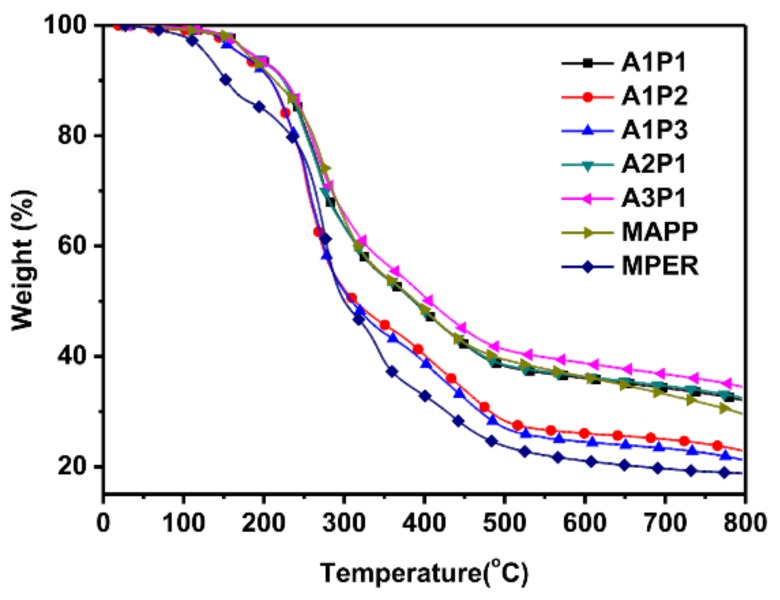
TGA curves of the adhesives.

**Figure 3 materials-13-00676-f003:**
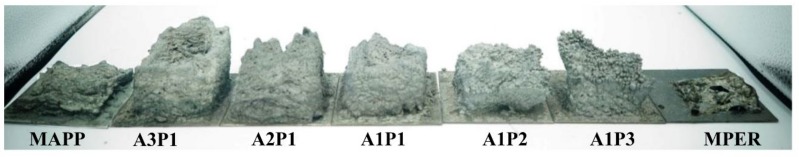
Digital images of intumescent char of adhesives.

**Figure 4 materials-13-00676-f004:**
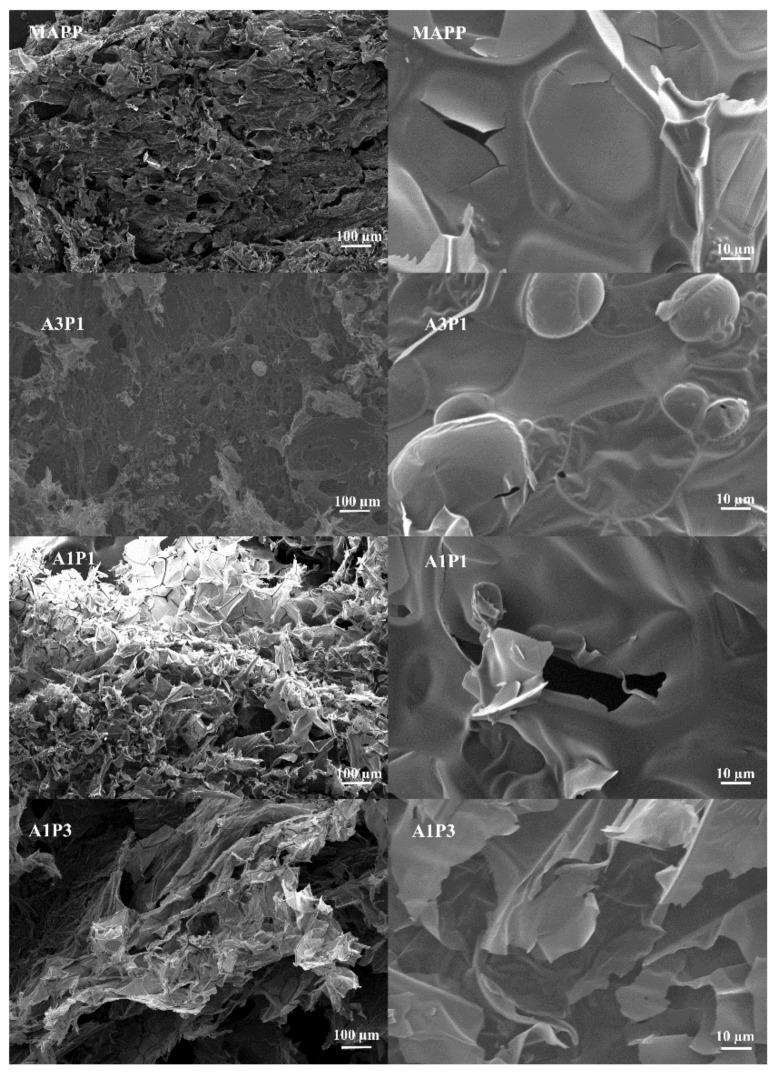
SEM images of flame retard resin char.

**Figure 5 materials-13-00676-f005:**
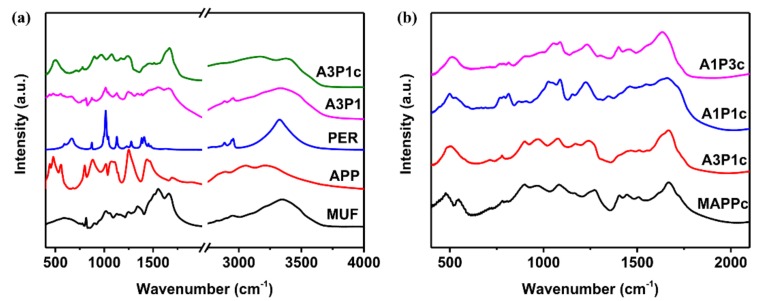
FTIR of the adhesive, its components (**a**) and char (**b**).

**Figure 6 materials-13-00676-f006:**
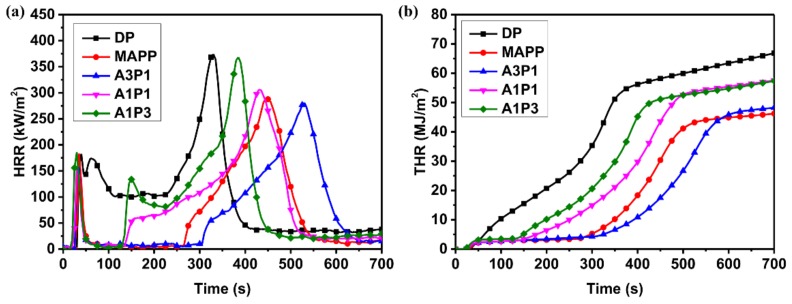
HRR (**a**) and THR (**b**) curves of the plywood samples.

**Figure 7 materials-13-00676-f007:**
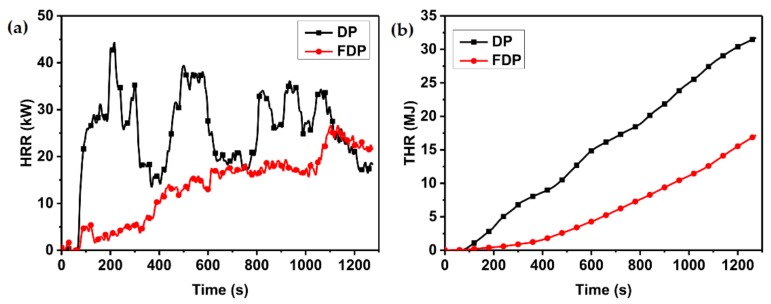
HRR (**a**) and THR (**b**) curves of the plywood samples provided by the SBI test.

**Figure 8 materials-13-00676-f008:**
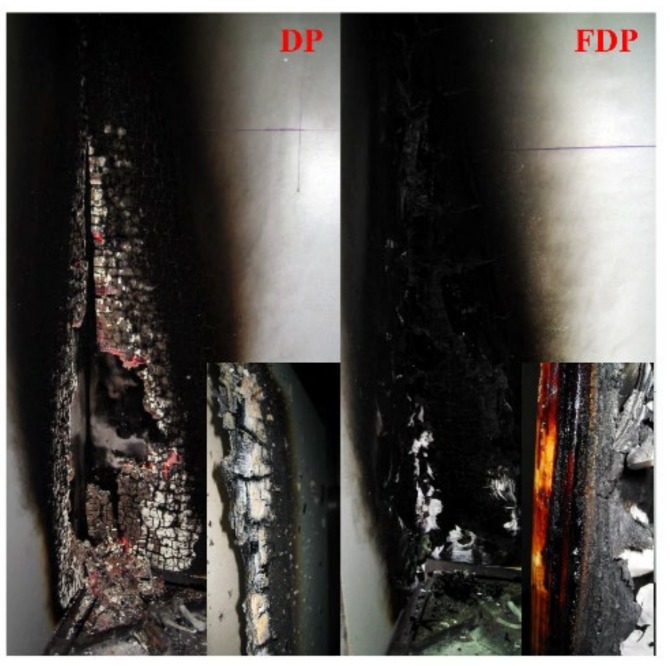
Images from the SBI test of plywood samples.

**Table 1 materials-13-00676-t001:** Formulation of resin samples.

Samples	MUF (g)	APP (g)	PER (g)
A1P3	100	10	30
A1P2	100	13.3	26.7
A1P1	100	20	20
A2P1	100	26.7	13.3
A3P1	100	30	10
MAPP	100	40	0
MPER	100	0	40
MUF	100	0	0

**Table 2 materials-13-00676-t002:** Criteria for compliance for classes involving SBI.

Class		Criteria for Compliance
A2		FIGRA_0.2MJ_ < 120 W/s;
		LFS < edge of specimen;
		THR_600s_ < 7.5 MJ
B1	B	FIGRA_0.2MJ_ < 120 W/s;
		LFS < edge of specimen;
		THR_600s_ < 7.5 MJ
	C	FIGRA_0.4MJ_ < 250 W/s;
		LFS < edge of specimen;
		THR_600s_ < 7.5 MJ
B2	D	FIGRA_0.4MJ_ < 750 W/s

**Table 3 materials-13-00676-t003:** Wet shear strength of the plywood with various ratios of MUF and IFR.

MUF:IFR (m/m)	100:0	100:40	100:60	100:80
Shear strength (MPa)	1.47 ± 0.02	1.35 ± 0.03	0.97 ± 0.07	0.62 ± 0.10

**Table 4 materials-13-00676-t004:** TGA Data of the adhesives.

Sample ID	T-5% (°C)	Residue at 800 °C (wt%)
MAPP	177.8	29.6%
A3P1	183.4	34.4%
A2P1	182.7	32.5%
A1P1	182.9	32.0%
A1P2	169.5	22.9%
A1P3	169.5	21.2%
MPER	127.3	18.7%

**Table 5 materials-13-00676-t005:** The data of the expansion volume of adhesives.

Sample ID	MAPP	A3P1	A2P1	A1P1	A1P2	A1P3	MPER
ex-V (mL)	36 ± 2	133 ± 4	89 ± 4	87 ± 3	76 ± 2	64 ± 2	17 ± 1

**Table 6 materials-13-00676-t006:** Cone calorimeter data of the plywood samples.

Sample ID	IT (s)	PHRR (kW/m^2^)			TTP^a^ (s)			AHHR (kW/m^2^)		THR (MJ/m^2^)	
		1st	2nd	3rd	1st	2nd	3rd	180 s	300 s	180 s	300 s
DP	26 ± 3	183 ± 6	174 ± 9	374 ± 10	40 ± 5	60 ± 10	330 ± 13	121 ±5	150 ± 10	19 ± 2	35 ± 3
MAPP	22 ± 2	183 ± 5	70 ± 5	289 ± 7	35 ± 4	294 ± 6	445 ± 9	18 ± 3	25 ± 3	3 ± 1	5 ± 1
A3P1	21 ± 2	158 ± 3	57 ± 3	281 ± 5	33 ± 4	325 ± 6	530 ± 8	19 ± 2	16 ± 2	3 ± 1	4 ± 1
A2P1	17 ± 2	151 ± 8	81 ± 11	297 ± 15	30 ± 8	278 ± 12	475 ± 16	18 ± 4	28 ± 5	3 ± 1	7±1
A1P1	18 ± 2	153 ± 5	59 ± 7	306 ± 8	33 ± 3	156 ± 6	430 ± 11	35 ± 3	57 ± 4	5 ± 1	15 ± 2
A1P2	17 ± 2	185 ± 4	114 ± 6	377 ± 5	30 ± 3	181 ± 5	405 ± 9	42 ± 3	69 ± 6	5 ± 2	18 ± 3

^a^ Time to PHRR.

**Table 7 materials-13-00676-t007:** SBI data of the plywood samples.

Parameters of SBI	DP	FDP
FIGRA_0.2MJ_	250.2	44.8
FIGRA_0.4MJ_	250.2	32.0
LFS < edge of specimen	Yes	Yes
THR_600s_	14.9	4.3

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
