# Peer review of "Preparation and Characterization of the Flame Retardant Decorated Plywood Based on the Intumescent Flame Retardant Adhesive"

_materials, 2020, doi:10.3390/ma13030676_

Round 1

Reviewer 1 Report

Dear authors.

After intensive reading of the manuscript i am satisfied with the quality of the paper and therefore i recommend the manuscript to be accepted by the editorial board in the present form.

I am very happy with processing of the experimental part of the manuscript, which was highly comprehensive and which is in good correlation with the content of introduction.

I want also thank you, for the opportunity to read an interesting article, which also increased mine knowledge of the proposed area of the research.

I wish you many successes in the future.

Author Response

We greatly appreciated the reviewer’s comments. We will try our best to do in this research direction with your encouragement. 

Reviewer 2 Report

Dear authors, only some suggestions:

-         Line 45 – 46: It is stated there that "Decorated panels are made from the common plywood and the decorative melamine-impregnated paper or wood veneers [11]." This is not entirely true, other construction materials are also used. Moreover, reference (11) is not at all related to that sentence.

-         Line 47: the abbreviations MDF and PB are not explained above.

-         Line 85: it is not correct to do the research with the hardener NH4Cl as it has been known for over 20 years that it is a carcinogen.

-         Line 93-94: what plywood was used and why such a small plywood dimension (100 x 100 mm) was used; this may significantly impair the parameters tested.

-         Line 94-99: please specify technical press conditions of veneering (pressure, time, temperature), otherwise it is not clear what samples were used. To be sure that the original plywood was not damaged by veneering.

-         Line 95-96: where suddenly a 18 mm thick plywood came from? If the plywood had previously been 9 mm thick and 2 veneer layers of 0.5 mm each were pressed onto it.

-         Line 144: Shear strength test is OK, what about bending strength test? Bending strength can say much more about the quality of the material produced.

-         Line 158: The values ​​given in Tab. 3 are not very high, commonly produced hardwood plywood have higher shear strength values. Has any very drastic exposure of the test specimens been used?

-         Line 294: It is stated there that "The flame-retardant class 293 of FDP with A3P1 was up to class B1(B), which was the highest level for wood materials." This is not entirely true, there are wood materials after retardation classified in the class "A".

Author Response

1. Line 45 – 46: It is stated there that "Decorated panels are made from the common plywood and the decorative melamine-impregnated paper or wood veneers [11]." This is not entirely true, other construction materials are also used. Moreover, reference (11) is not at all related to that sentence.

Answer: We greatly appreciated the reviewer’s question. We agree with the reviewer that the sentence is not accurate. Thus, the corresponding sentence have been changed to “Decorated panels are mainly  made from the common panels and the covering materials that can be paints, veneer, laminating plastics or impregnated papers. Many laminated wood panels are covered with the resin impregnated papers [11].  [11]. ” (11, A. Kandelbauer, P. Petek, S. Medved, A. Pizzi, A. Teischinger, On the performance of a melamine–urea–formaldehyde resin for decorative paper coatings, European Journal of Wood and Wood Products 68(1) (2010) 63-75. )

2. Line 47: the abbreviations MDF and PB are not explained above.

Answer: Thanks a lot for the reviewer’s good reminder. The full names of MDF and PB have been added into the manuscript. The corresponding sentence have been changed to “Statistical results show that more than 90% of medium density fiberboard (MDF) and particleboard (PB), and 60% of plywood are overlaid with the impregnated paper or the decorative veneer in China.”

3. Line 85: it is not correct to do the research with the hardener NH4Cl as it has been known for over 20 years that it is a carcinogen.

Answer: We greatly appreciated the reviewer’s excellent reminder and recommendation. NH4Cl is used to be a weak acid to adjust the pH value in the synthesis progress of MUF resin.  Considering NH4Cl is a carcinogen, it can be taken place in the real and big scale production. Many weak acids can be a candidate. For example, formic acid.

4. Line 93-94: what plywood was used and why such a small plywood dimension (100 x 100 mm) was used; this may significantly impair the parameters tested.

Answer: Thanks a lot for the reviewer’s important question. There are some mistakes and insufficient for the description of the plywood we used. The plywood with the 9 mm thickness is 7-layer poplar plywood. The dimension in the manuscript of the plywood is of the samples for CONE, not their prepared dimensions. Thus, the corresponding sentences have been change to “The decorated plywood was prepared through two steps. Firstly, the flame retardant adhesive resin was coated on the surface of poplar plywood (300 mm × 300 mm × 9 mm). Secondly, a piece of decorative veneer (310 mm×310 mm×0.5 mm) was hot-pressed on the top of the coated plywood with 8 MPa and 110 oC and kept for 5 min. The specimen for CONE was cut into 100 mm × 100 mm. FRD-plywood (a big scale sample for SBI) was prepared based on A3P1 adhesive and plywood (1220 mm ×2440 mm ×18 mm). ”.

5. Line 94-99: please specify technical press conditions of veneering (pressure, time, temperature), otherwise it is not clear what samples were used. To be sure that the original plywood was not damaged by veneering.

Answer: Thanks a lot for the reviewer’s question. The veneering pressure is 8 MPa and the time and temperature are 5 min and 110 oC, respectively. Due to the lower temperature and shorter relatively time, the original plywood might not be damaged by veneering. Thus, the corresponding sentences have been change to “Secondly, a piece of decorative veneer (310 mm×310 mm×0.5 mm) was hot-pressed on the top of the coated plywood with 8 MPa and 110 oC and kept for 5 min. The specimen for CONE was cut into 100 mm × 100 mm. FRD-plywood (a big scale sample for SBI) was prepared based on A3P1 adhesive and plywood (1220 mm ×2440 mm ×18 mm).”.

6. Line 95-96: where suddenly a 18 mm thick plywood came from? If the plywood had previously been 9 mm thick and 2 veneer layers of 0.5 mm each were pressed onto it.

Answer: We greatly appreciated the reviewer’s reminder and question. The 18 mm thick plywood was a kind of big scale samples which was prepared for SBI test with 1220 mm ×2440 mm ×18 mm.  The 18 mm plywood was applied widely in the furniture manufacture and the indoor decoration.

7. Line 144: Shear strength test is OK, what about bending strength test? Bending strength can say much more about the quality of the material produced.

Answer: We greatly appreciated the reviewer’s recommendation and question. We agree with the reviewer that the bending strength can say more. In the manuscript, the shear strength of the plywood with the flame-retardant resin was measured to indicate the bonding strength of the flame-retardant adhesive. The flame-retardant adhesive is only used between the decorative veneer and the plywood substrate which was prepared by the normal adhesive. So the influence of the flame-retardant adhesive on the quality of the plywood is limited.

8. Line 158: The values ​​given in Tab. 3 are not very high, commonly produced hardwood plywood have higher shear strength values. Has any very drastic exposure of the test specimens been used?

Answer: We greatly appreciated the reviewer’s question. The veneers we used are poplar without the drastic exposure. After the hot-press, the samples are not exposed drastically, either.

9. Line 294: It is stated there that "The flame-retardant class 293 of FDP with A3P1 was up to class B1(B), which was the highest level for wood materials." This is not entirely true, there are wood materials after retardation classified in the class "A".

Answer: We greatly appreciated the reviewer’s excellent reminder and recommendation. The content of “the highest level for wood materials” is not correct. The corresponding sentence has been deleted.

Reviewer 3 Report

The authors have reported on the preparation and characterization of the flame-retardant decorated plywood based on the intumescent flame retardant adhesive. The paper presents an interesting piece of work and will be useful to wide audiences. I recommend its acceptance for publication with monor revision, as below.

Some specifications for the decorative veneers and plywood should be provided. The real fire takes place in the air or ambient atmosphere. It is desirable that the authors do TGA in air atmosphere along with in nitrogen atmosphere.

Author Response

Some specifications for the decorative veneers and plywood should be provided. The real fire takes place in the air or ambient atmosphere. It is desirable that the authors do TGA in air atmosphere along with in nitrogen atmosphere.

Answer: We greatly appreciated the reviewer’s comments and recommendation. We totally agree with the reviewer that the real fire takes place in the air or ambient atmosphere. TGA in air atmosphere is closer the real fire condition than in inert atmosphere. TGA in air atmosphere was also measured and gave us the similar results with that of TGA in inert atmosphere. The reason may be related to the process of char formation. The earliest part of char could barrier oxygen in the next formation of char.

Reviewer 4 Report

The authors report on the preparation and testing of flame-retardant plywood. The work is well designed and performed and can be accepted in its present form.

Author Response

We greatly appreciated the reviewer’s comments. We will try our best to do in this research direction.